# Acceptability of Healthcare Professionals to Get Vaccinated against COVID-19 Two Weeks before Initiation of National Vaccination

**DOI:** 10.3390/medicina57060611

**Published:** 2021-06-12

**Authors:** Athanasia Pataka, Seraphim Kotoulas, Emilia Stefanidou, Ioanna Grigoriou, Asterios Tzinas, Ioanna Tsiouprou, Paul Zarogoulidis, Nikolaos Courcoutsakis, Paraskevi Argyropoulou

**Affiliations:** 1Respiratory Failure Unit, G Papanikolaou Hospital, Aristotle University of Thessaloniki, 55236 Thessaloniki, Greece; patakath@yahoo.gr (A.P.); akiskotoulas@hotmail.com (S.K.); a_stefanidou@hotmail.com (E.S.); ioagrig@hotmail.gr (I.G.); stergiostzinas@hotmail.com (A.T.); pargyrop@auth.gr (P.A.); 2Department of Respiratory Medicine, G Papanikolaou Hospital, Aristotle University of Thessaloniki, 55236 Thessaloniki, Greece; joanna_tsi@hotmail.com; 33rd University General Hospital, “AHEPA” University Hospital, 55236 Thessaloniki, Greece; 4Pulmonary Department, “Bioclinic” Private Hospital, 55236 Thessaloniki, Greece; 5Radiology Department, Democritus University of Thrace, 68100 Alexandroupolis, Greece; ncoucourcou@med.duth.gr

**Keywords:** vaccination, acceptance, SARS-CoV-2, COVID-19, healthcare professional, physicians, nurses, Greece

## Abstract

*Background and Objectives* The greatest challenge vaccines face is that of acceptance from the general population. Healthcare professionals’ (HCPs) recommendations have significant influence on general public vaccination behavior. The aim of this study was to assess the willingness of HCPs to get vaccinated against COVID-19, two weeks before initiation of vaccinations. *Materials and Methods:* We conducted an anonymous online survey from 11–15 December 2020 among HCPs by emails delivered from the local medical and nursing stuff associations. *Results:* The 71.1% of 656 HCPs intended to accept vaccination, 5.9% did not and 23% were still undecided. The acceptance rate was higher in physicians (76.5%) and significantly lower in nurses (48.3%). Most of the responders who intended to accept vaccination were males (*p* = 0.01), physicians (*p* = 0.001), older (*p* = 0.02), married (*p* = 0.054) with children (*p* = 0.001), and had treated patients with COVID-19 (*p* < 0.001). In the multivariate logistic regression, the predictors of HCPs willingness to get vaccinated were parenthood (OR = 4.19, *p* = 0.003), being a physician (OR = 2.79, *p* = 0.04), and treating confirmed/suspected COVID-19 patients (OR = 2.87, *p* = 0.036). *Conclusions:* Low vaccination acceptance rate was found especially in nurses, and as this may have a negative impact in the vaccination compliance of the general public, interventional educational programs to enhance vaccination are crucial.

## 1. Introduction

At the end of 2019, a novel highly infectious corona virus (SARS-CoV-2, COVID-19) causing serious acute respiratory syndrome emerged in Wuhan, China, and rapidly spread to other countries all over the world. The World Health Organization (WHO) declared COVID-19 a pandemic on March 2020, with millions of people having been infected worldwide since that time [1]. The COVID-19 pandemic has caused a negative impact on people’s health, psychology, behavior, and quality of life and has also affected economies around the world. Strategies, such as lock downs with closure of shopping centers, educational establishments, restaurants, bars, curfews, social distancing, isolation of positive cases, and practices for shielding the elderly have been used in order to slow down the transmission of COVID-19. Despite these strategies, the death toll is increasing, and the need of the presence of an effective vaccine is crucial.

The development of COVID-19 vaccines is a challenge. During the last months massive vaccinations have begun worldwide. The European Medicines Agency (EMA) and the European Commission has authorized Pfizer-BioNTech COVID-19 vaccine, Moderna COVID-19 vaccine, COVID-19 Vaccine AstraZeneca and COVID-19 Vaccine Janssen in European countries [2]. At the same time, several laboratories and institutions are working to obtain COVID-19 vaccines with many ongoing clinical trials. However, the greatest challenge vaccines face after their development is that of acceptance, seeing as vaccine hesitancy has been a barrier to fully vaccinate the general population even prior to the COVID-19 pandemic. The effectiveness of vaccines depends on their uptake by the population. Vaccine uptake may be influenced by concerns about their safety and effectiveness, low risk perception, lack of recommendation and information by health professionals, and socioeconomic factors. Vaccine hesitancy constitutes a growing concern and includes the concerns people express by refusing or delaying vaccines despite their availability [3,4,5]. The World Health Organization (WHO) considered vaccine hesitancy to be one of the top ten health threats globally [6].

Healthcare professionals’ (HCP) attitude toward the vaccine and their intention to use and to recommend it to their patients play a determinant role. It has been reported that healthcare professionals with unfavorable attitudes or hesitations toward vaccinations may transmit these hostile attitudes to their patients and recommend vaccination less frequently [7]. Patients rely on the opinion of HCPs about vaccination. HCPs that are confident for vaccinations and provide easily understandable educational information have been shown to be useful in improving patient acceptance and decisions about immunization [8,9,10].

During the last months, COVID-19 vaccination began in Greece, with HCPs being prioritized. The aim of this study was to assess the willingness of HCPs to get vaccinated against COVID-19, two weeks before the initiation of vaccinations.

## 2. Methods

We conducted an anonymous online survey from the 11th to the 15th of December 2020 among HCPs of Northern Greece by emails delivered from the local medical and nursing stuff associations. The total number of affiliated was estimated to 8800. The northern part of Greece was more severely affected by COVID-19 during the second wave of the pandemic.

Information was collected through an anonymous online survey and structured questionnaire. An information sheet from the research team was included in the survey with a detailed explanation of the research purpose. The participants were not able to continue to the next question of the questionnaire if they failed to provide a response to an item. The participants were informed about the use and anonymization of the data and that their anonymity would be perceived. No incentives were offered to the participants of the survey.

The questionnaire addressed: (1) sociodemographic and general characteristics of the HCP (age, gender, marital and parenthood status, and change of financial status during the pandemic); (2) occupation (physician, nurse, technician, or other healthcare professionals), location of work (e.g., hospital or private practice), medical discipline (e.g., internal medicine, surgery, laboratory/imagine or intensive care unit), years of experience, educational level (MD, phD), whether they directly provided care, diagnosed, and treated patients with suspected or confirmed SARS-CoV-2; (3) the risk perception about COVID-19 was examined with the following questions: ‘Do you feel at risk of getting infected with COVID-19′, ‘How vulnerable do you consider yourself to a COVID-19 infection?’ and ‘If you were infected with COVID-19, how serious would your condition be?’, rated on a 7-point Likert scale from 1, indicating very mild, to 7, indicating extremely; (4) prior infection of COVID-19; and (5) willingness to get vaccinated with COVID-19 vaccine with a 5-point Likert scale (1 = strongly disagree, 2 = disagree, 3 = don’t know, 4 = agree, and 5 = strongly agree). The estimated Cronbach’s alpha in ‘Do you feel at risk of getting infected with COVID-19′ was 0.87; in ‘How vulnerable do you consider yourself to a COVID-19 infection?’, it was 0.82; in ‘If you were infected with COVID-19, how serious would your condition be?’, it was 0.76; and in willingness to get vaccinated with COVID-19 vaccine, it was 0.94.

The protocol was approved by the institutional review board (N 876, Approval Date 20 May 2020) of G Papanikolaou Hospital, Exohi, Thessaloniki, Greece, and all the responders gave their written electronic consent form. A pilot study was conducted before the actual data collection from a sample of 30 healthcare professionals working in G Papanikolaou Hospital who completed the questionnaire, and the results were not included in the survey.

## 3. Statistical Analysis

Data were analyzed using IBM SPSS version 21.0 (IBM Corporation, Armonk, NY, USA). Continuous variables were presented as mean ± SD unless otherwise stated. Absolute and relative frequencies were used for categorical variables. Analysis of variance (ANOVA) was used for the analysis of the differences between group means. Tests were two-tailed, and *p* < 0.05 was accepted as statistically significant. The sample size was estimated with prevalence of participants willing to receive vaccination against COVID-19 set at 50%, a 95% confidence interval, and a relative precision of 5%. The estimated minimum sample size was 490. We combined survey responses into two categories (strongly agree/agree; don’t know/disagree/strongly disagree). The difference between groups with intention or not to accept vaccination and those undecided was examined using Chi-square test. Univariate logistic regression models were performed to assess relations between the different variables (gender, age, marital status, and parenthood), specialty (surgical vs. internal medicine, laboratory vs. internal medicine), ICU work setting, type of HCP (i.e., physicians vs. nurses), work with confirmed/suspected COVID-19, infection of COVID-19, and the participants’ willingness to receive vaccination.

In order to isolate predictors of HCP willingness to get vaccinated against COVID-19, we performed a multivariate logistic regression. The following independent variables were included: age in years (continuous), gender (male = 0; female = 1), marital (single/divorced/widowed = 0; married = 1) and parental status (no children = 0, having children = 1), type of healthcare worker (1 = doctors, 2 = nurses, 3 = others as technicians, laboratory, and physiotherapists), specialty (internal medicine = 1, surgical = 2, laboratory/imaging = 3, ICU = 4), treating patients with COVID-19 infection (no = 0, yes = 1) and being infected with COVID-19 themselves (no = 0, yes = 1). The odds ratios (OR) were presented with a 95% confidence interval (CI).

## 4. Results

A total of 656 HCPs completed the online questionnaire. It is estimated that around 8800 were invited from the affiliated members of the associations (response rate 7.5%). Most of the responders were females (55.9%), aged 44.8 ± 12.4 years, and married (64%) with children (63.8%) (Table 1). Regarding their work-related characteristics, the majority of the responders were physicians (78%), 58.4% worked in the private sector, 48.6% treated suspected or confirmed COVID-19 patients, and 13% worked in an ICU. Moreover, 10% of the responders were confirmed of have been infected with COVID-19, 29.7% thought they were at extreme risk to be infected with COVID-19, and 18.9% thought their condition would be severe if infected (Table 1).

Of the responders, 71.1% intended to accept vaccination for COVID-19, whereas 5.9% did not intend to, and 23% were still undecided. The acceptance rate was 76.5% in physicians and significantly lower, 48.3%, in nurses. The differences between the responders according to their intention to accept COVID-19 vaccination are presented in Table 2.

Most of the responders who intended to accept vaccination were males (*p* = 0.01), physicians (*p* = 0.001), older (*p* = 0.02), married (*p* = 0.054) with children (*p* = 0.001), and had treated patients with suspected or confirmed COVID-19 (*p* < 0.001). However, 50.7% of those who indented to be vaccinated expressed fear for possible side effects. Most of the undecided responders and those who did not intend to accept were women (*p* = 0.01), nurses (*p* = 0.001), and single with no children (*p* = 0.001). No significant differences were found toward the acceptance of the vaccine between medical specialties or location of work, apart from those working in the ICU, where 14.3% did not intend to or were undecided (mostly nursing stuff) (*p* = 0.01). Prior infection with COVID-19 (*p* = 0.57), feeling at risk of getting infected (*p* = 0.49), and vulnerable if infected (*p* = 0.2) did not differ between groups.

In order to investigate possible predictors of HCPs willingness to get vaccinated, we performed a multivariate logistic regression with age, gender, marital and parental status, type of healthcare worker, specialty (internal medicine, surgical, laboratory, imaging, and ICU), treating patients with COVID-19 infection and prior infection of COVID-19 as variables (Table 3). In the univariate analysis, the factors associated with intentions to accept COVID-19 vaccination were male gender (odds ratio, OR = 1.89, *p* = 0.018), age (OR = 0.97, *p* = 0.017), children (OR = 2.249, *p* = 0.004), and being a physician (OR = 3.49, *p* = 0.002) (Table 3).

However, in the adjusted model, the only factors that remained significant were parenthood (OR = 4.19, *p* = 0.003), being a physician (OR = 2.79, *p* = 0.04), and treating confirmed/suspected COVID-19 patients (OR = 2.87, *p* = 0.036). Worse economic situation during the pandemic did not affect the results (OR = 1.12, *p* = 0.82).

## 5. Discussion

COVID-19 has rapidly spread due to its high contagion rate affecting all the countries in the world with great social and economic impact. Several vaccines have been developed or are under development globally, but on the other hand, this rapid development raises concerns about their safety and may contribute to vaccine hesitancy [11]. Someone would expect that the great impact of the pandemic on the overburdened healthcare system would result in a positive and massive acceptance of the vaccine, especially among the HCPs. Interestingly our data demonstrated that 71.1% of HCPs intended to accept vaccination for COVID-19, whereas 5.9% did not intend to, and 23% were still undecided even a few days before the initiation of national vaccination. Positive predictors for vaccination included occupation as physician, parenthood, and treating patients with suspected or confirmed COVID-19, whereas negative predictors included occupation as nursing stuff.

The effectiveness of vaccination highly relies on the uptake rate of the vaccine from the general population. One of the major obstacles to combating the COVID-19 pandemic is hesitancy for receiving the vaccine, and as it has been recently reported, there are increasing reports on COVID-19 vaccination hesitancy among the general population in European countries [12]. There are important differences in the percentage of individuals that are willing to vaccinate or that present a positive intention toward COVID-19 vaccine between countries varying between 37.5% and 89%. [11] Data from a study performed in the Greek general population before the development of COVID-19 vaccines (May 2020) showed that 57.7% were willing to get vaccinated, whereas 40% were unwilling or unsure to receive it [13]. Participants > 65 years old, with a family member or themselves belonging to a vulnerable group and those with greater knowledge about COVID-19 (transmission, prevention, and control measures) were more likely to accept vaccination [13]. Data from Australia estimated that almost 14.3% of people would refuse or were undecided about receiving a COVID-19 vaccine, with lower education level and inadequate health literacy being associated with vaccine reluctance [14]. Similarly, data from USA estimated a 20% refusal rate, with 70% of responders concerned about vaccine safety, 42% about developing COVID-19 from the vaccine, and 30% about vaccine effectiveness [15]. Additionally, in France, 26% of people responded that they would refuse a COVID-19 vaccine, especially those with lower incomes and women aged < 35 years and aged > 75 years [16]. In a study from Chile, where the infection rate was rather high, 90.6% of the responders were willing to pay for a COVID-19 vaccine, whereas 92.4% believed that they would be infected with COVID-19 [17]. Recent data from Malta, reported that over 50% of participants were willing to take the vaccine, with males being more willing. Vaccine hesitancy resulted with 32.6% of the study population being unsure and 15.6% declaring not willing to be vaccinated. Females were more unsure. Lack of data for vaccine safety has been reported as the main reason for unwillingness [18]. A national survey in the United States revealed that 42.4% of participants were hesitant to be vaccinated against COVID-19, and approximately 1 in 10 did not intend to be vaccinated. The strongest independent predictors of being hesitant were lower educational attainment, Black race, perceived personal risk for COVID-19, and not having been recently vaccinated for influenza [19].

Regional differences can be crucial in addressing and fighting the problem of vaccine hesitancy. There are data that people in Eastern Europe are the least likely to agree that vaccines are effective, as opposed to South Asia or Eastern Africa [20]. In the general public, more pronounced low COVID-19 vaccine acceptance rates were reported in the Middle East, Eastern Europe, and Russia, whereas high acceptance rates were reported in East and South East Asia. Considerable difference in the willingness of the general population to get vaccinated has been described between Asian and European countries, where the countries that exceeded 80% of positive response tended to be Asian [12]. Differences between countries on vaccination acceptance reflect individual preference, sociodemographic factors, perceived health risk depending on COVID-19 prevalence, intensity, mortality, and impact on the healthcare system. Additionally, the restrictive preventive public health measures, previous experience with other vaccines, and educational programs for the general population play an important role. There is evidence that indicates that education, information, and communication improves the willingness of the public to vaccinate. In a survey in the personnel and the students of a university in Southern Italy, 84.1% were willing to receive a future COVID-19 vaccine, although some concerns have been expressed regarding its safety and effectiveness. Only 21.4% of respondents were not worried about the safety of the vaccine at all [21].

HCPs’ role is very important in this setting. The willingness of HCPs to be vaccinated against COVID-19 also varies among countries. For example, Congolese HCPs (27.7%) presented a very low acceptance rate compared with French HCP (77.6%) [22,23,24]. However in a recent study, the majority of healthcare workers in Asia (more than 95%) were willing to receive COVID-19 vaccination. The acceptance was greater in those that had the perception that the pandemic was severe, that the vaccine was safe and effective, those that expressed altruism, had fewer financial concerns, and those that trusted the healthcare authorities [25]. In a systematic review of vaccine acceptance rates among healthcare workers (doctors and nurses), the vaccine acceptance rates ranged from low (27.7%) in the Democratic Republic of the Congo to high (78.1%) in Israel [26]. Additionally, another review reported that the prevalence of vaccination hesitancy worldwide for COVID-19 in HCP ranged from 4.3% to 72%, with the most important factors affecting acceptance being the concerns about vaccine efficacy, safety, and potential side effects. This review found that males, of older age, with higher educational degree (i.e., doctoral degree holders), physicians, with history of influenza vaccination, treating patients suffering from COVID-19, and with higher perceived risk of getting infected were more likely to accept COVID-19 vaccination [27]. On the other hand, in a report from Taiwan, only 23.4% of HCP and 30.7% of outpatients were willing to receive vaccination, possibly due to the relatively safe status of COVID-19 in Taiwan [28]. By contrast, in China, 75% of the doctors and 68% of the nurses intended to take up free COVID-19 vaccination, which dropped to 64.6% for doctors and 56.5% for nurses if a fee was applied in the vaccine [29]. In a similar study in China, the acceptance rate was 72.7%/71.2% if the vaccination was recommended by the government/hospitals; however, it dropped to <50% if even mild side effects were common [30]. In another study, in Zhejiang Province, China, 79% of HCPs expressed their willingness to get vaccinated with those < 50 years old, and those who believed that they were likely to get infected with COVID-19 were more willing to get vaccinated [31].

HCPs are the ones at the forefront of COVID-19 vaccine delivery and also are the initial target groups for COVID-19 vaccine receipt. As the COVID-19 vaccination began, the Greek government included HCPs of the public and private sector into a free COVID-19 vaccination program and prioritized those working in high-risk settings, as they are exposed to greater risk. Our results indicate that the rate of acceptance of COVID-19 vaccine was higher in HCPs working with COVID-19 patients and also among physicians (76.5%) compared with nurses (48.3%) and other HCPs. This is in accordance with other studies from other countries [24,31,32,33]. A previous study in Greece suggested that willingness to get vaccinated ranged around 43% among HCP [34] A questionnaire-based survey in France and in French-speaking parts of Belgium and Canada reported that 72.4% of the participating HCPs would be in favor of getting vaccinated for COVID-19, and 79.6% would recommend it to their patients [35]. Additional data from French HCP report that 76.9% would accept COVID-19 vaccination. Older age, male gender, fear about COVID-19, individual perceived risk, and history of flu vaccination were associated with vaccine acceptance. Vaccine hesitancy was related with decreased COVID-19 vaccination acceptance. Nurses and assistant nurses were less prone to accept vaccination compared with physicians [24].

Low vaccination acceptance rate among nurses has also been observed in other studies. In a study in Congo [22], 24% of nurses and other HCPs vs. 37.7% of doctors intended to accept COVID-19 vaccine, whereas an Israeli study showed 61% of nurses vs. 78% of doctors [32,33]. In a study in Hong Kong, 40.0% of nurses intended to accept vaccination. Higher intention to accept was observed in those working in private sector, treating suspected or confirmed COVID-19 patients, those with chronic conditions, and those who accepted influenza vaccination in 2019 [36]. Another study in Hong Kong, which evaluated the proportion of nurses (90% females) who intended to take COVID-19 vaccine and also the influenza vaccine uptake rate, found 63% and 49% acceptance rates, respectively. Factors associated with stronger intention to be vaccinated against COVID-19 were younger age, stronger confidence, weaker complacency, and collective responsibility [37]. Low vaccination acceptance rate among nurses may have a negative impact in future vaccination compliance of people who relate professionally or personally with vaccine hesitant nurses or other HCPs. Interventional educational programs to enhance vaccination among hesitant HCPs are crucial.

A positive association was observed between male gender and intention of HCPs to accept COVID-19 vaccination, as in other previous studies [18,23,24,38]. The decision-making process of vaccination acceptance may be influenced by the perceptions of risk from COVID-19. The data suggest that there is a predominance of male gender in mortality rate and complications from COVID-19 [36]. Additionally, comorbidities such as chronic respiratory disease and cardiovascular disease are more frequent in men, and this could bias males to accept the vaccination more easily [24]. Additionally, we found that older HCPs presented a higher acceptance vaccination rate. This may be due to the fact that apart from people with serious comorbidities, older adults are particularly vulnerable to worse outcomes from COVID-19, creating fear among the elderly and thus a higher vaccination acceptance [22,23,24]. In addition, married responders with children presented higher acceptance rate; however, parenthood remained a significant predictive factor after multivariate analysis, possibly indicating the responsibility of parents toward their children especially in HCPs treating COVID-19 patients. In our study, no significant differences were observed between specialties (internal medicine, surgical, laboratory, and imaging). Other studies have found that HCPs working in internal medicine departments displayed significantly higher vaccination acceptance than those working in general surgery departments [32].

The main limitation of this study is the potential selection bias in the sample. The majority of the HCPs of the study were physicians (78%), with only 17.5% nurses and 4.5% of other HCPs, such as physiotherapists and technicians. The HCPs were invited by the assistance of the local medical and nursing stuff associations who sent the e-mails with the questionnaire to all their participants. However, only 656 of them answered the questionnaire. The response rate was rather low, reaching 7.5%. A limitation of the questionnaire was that the participants were not able to continue to the next question if they failed to provide a response to an item; therefore, some may not have completed the questionnaire due to that reason. Additionally, the representativeness of the sample, the recall bias, and possible social desirability bias could be considered as limitations of the study. At that time, northern Greece was more severely affected by COVID-19 compared to the other areas. The study design could be improved in a future project including a greater number of participants from more than one region (ideally national) and several associations, with a greater response rate of all the types of HCPs. As human memory is frequently imprecise, recall bias could be included in the limitations of the study. Social desirability could also be considered a limitation, as the participants might have tended to over-report the more socially desirable answers and to under-report their actual attitudes. Due to the low response rate, the responders might have been more concerned about the pandemic situation than those who did not participate, and possibly, the vaccination acceptance rate could be lower than what was found in our study. On the other hand, we recruited adequate number of HCPs from both genders, from both public and private sector. Unfortunately, we did not have data from the general population at that time in order to compare, and we also had inadequate data for comorbidities and prior acceptance of other vaccines as influenza and H1N1 [36] that may be indicative of the behavior toward vaccine acceptability. However, there are data reporting that Greeks presented lower acceptance rates of H1N1 vaccination (9.1 to 22.9%) comparing to other European populations [39,40].

HCPs’ recommendations have a significant influence on their patients’ vaccination behavior as they serve as a source of information for the general public. COVID-19 vaccination acceptance could lead to reduced morbidity and mortality releasing the overwhelming healthcare resources [41,42]. HCPs’ consultation is a significant key factor in patients’ decision to accept vaccination. As we observed a lower acceptability of vaccination in nursing stuff participants, we believe that there is a significant need for increasing awareness and education in order to improve higher acceptance of COVID-19 vaccination. Otherwise, there is a chance of a high rejection rate in the general population. During the pandemic, misleading media reports and conspiracy theories may reduce confidence of vaccination in the general public but also in HCPs [43]. Additionally, conflicts of interests and controversies between experts and companies may also discourage vaccination [40,41,42]. A national evidence-based strategy is essential in order to effectively promote the uptake of vaccination for HCPs, not only for their protection but also for reducing transmission to their families and community. Furthermore, HCPs serve as role models for their patients and the whole community. As massive vaccination recently started, it is very important for governments to disseminate information and evidence of its efficacy and safety in order to dissolve all possible hesitations [42,43,44,45].

It is important to establish the appropriate strategies in order to improve vaccination compliance among HCPs. An initial step for building trust toward COVID-19 vaccination should aim toward the resolution of vaccine hesitancy in various countries [44]. This should include the efforts of health policy makers, governments, and media (i.e., including social media). COVID-19 vaccination trust among the general public may be achieved through the spread of clear messages through independent committees and trusted bodies (as national immunization committees), even trusted media advocating the safety and efficacy of currently available vaccines. In this respect, HCPs should be informed with credible information from national committees and professional associations about vaccines’ safety and efficacy. The monitoring of side effects and the regular reactive feedback to HCPs could ensure trust in both vaccines and health authorities. Additionally, addressing HCPs’ vaccination concerns is essential for the resolution of vaccine hesitancy. The upcoming vaccination campaigns should provide effective interventions targeting HCPs to resolve their general and specific vaccine hesitancy in order to build trust in the general population [41,42].

## Figures and Tables

**Table 1 medicina-57-00611-t001:** Characteristics of the responders.

Variable	Frequency % (*n* = 656)
Sex (Males/Females)	44.1%/55.9%
Age (years)	44.8 ± 12.4
Marital status	
married	64%
divorced	6.10%
widowed	0.90%
single	29%
Children	
Yes	63.80%
Years of experience	
<5	16.70%
>20	24.90%
11–15	16.80%
16–20	19.60%
5–10	22%
Occupation	
Doctors	78%
Nurses	17.50%
Other	4.50%
Specialty	
Internal medicine	54%
Surgical	35%
Laboratory, radiology	11%
Working with COVID-19 patients	
Yes	48.60%
No	38.80%
Don’t know	12.60%
Working in the ICU	13%
Financial status after COVID-19 pandemic	
Same	31.60%
Worse	41.20%
Significantly worse	27.30%
No answer	14.60%
Infected from COVID-19	
Yes	10.60%
No	76.80%
Don’t know	12.60%
Do you feel at risk of getting infected with COVID?	
1 not at all	1.30%
2	4.50%
3	8.70%
4	18.30%
5	21.20%
6	18.90%
7 extremely	27.10%
How vulnerable do you consider yourself to a COVID-19 infection?	
1 not at all	9.80%
2	17.00%
3	10.70%
4	22.20%
5	15.30%
6	14.30%
7 extremely venerable	10.70%
If you were infected with COVID-19, how serious would your condition be?	
1 very mild	3.60%
2	10.40%
3	13%
4	22.70%
5	18.10%
6	13.60%
7 extremely severe	18.60%

**Table 2 medicina-57-00611-t002:** Differences of the responders according to their intention to accept COVID-19 vaccination.

	Intend to Accept Vaccination	Not Intend to Accept	Undecided	*p*
age	45.25 ± 12.8	37.6 ± 11.2	42.5 ± 11.9	0.02 *
gender (m/f)	78.7%/66.1%	4.1%/7.1%	17.2%/26.8%	0.01
marital status				
single	62.50%	7.30%	30.20%	0.054
married	75.80%	4.60%	19.60%	
children yes	79.40%	3.20%	17.30%	0.001
children no	58%	10%	32%	
work setting				
public	77%	5.40%	17.50%	0.1
private	76.90%	4.20%	18.80%	
specialty				
internal medicine	75%	3.90%	21%	NS
surgical	80.8%	2.80%	16.40%	
laboratory	76.20%	4.70%	19%	
ICU	71.40%	14.3%^	14.30%	
Treat confirmed/suspected COVID	73%	7%	19.90%	<0.001
physicians/nurses	76.5%/48.3%	4%/13.8%	19.4%/37.9%	0.001
financial status				
Same	74.70%	3.40%	21.80%	0.07
Worse	68%	10.90%	21%	
Significantly worse	71.70%	2%	26.20%	

* post hoc: age intent to accept vs. not intent *p* = 0.04, ^ *p* = 0.01.

**Table 3 medicina-57-00611-t003:** Factors associated with intentions to accept COVID-19 vaccination.

	Univariate OR, (95% CI)	*p*	Multivariate ^ OR, (95% CI)	*p*
gender (male)	1.9 (1.11–3.21)	0.018	0.98 (0.52–1.85)	0.96
age	0.97 (0.95–0.995)	0.017	1.008 (0.98–1.034)	0.55
married	1.3 (0.76–2.34)	0.31	1.78 (0.68–4.7)	0.24
parenthood	2.25 (1.3–3.89)	0.004	4.19 (1.63–10.75)	0.003
specialty				
surgical vs. internal medicine	0.7 (0.35–1.4)	0.7	0.75 (0.36–1.55)	0.4
laboratory vs. internal medicine	0.93 (0.32–2.7)	0.93	0.825 (0.29–1.85)	0.74
ICU work setting	1.17 (0.49–2.78)	0.7	1.36 (0.54–3.45)	0.5
physicians vs. nurses	3.49 (1.59–7.66)	0.002	2.79 (1.27–6.89)	0.04
treat confirmed/suspected COVID	1.45 (0.645–3.32)	0.3	2.87 (1.07–7.735)	0.036
Infected from COVID	1.24 (0.54–2.88)	0.6	1.48 (0.46–4.78)	0.5

OR = odds ratio, CI = confidential interval, ^age, gender, marital status, children, specialty, work in ICU setting, and post infection of COVID-19.

## Data Availability

Data available on request due to restrictions of privacy of the participants. The data presented in this study are available on request from the corresponding author. The data are not publicly available due to the initial design of the study.

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
