# Peer review of "Acceptability of Healthcare Professionals to Get Vaccinated against COVID-19 Two Weeks before Initiation of National Vaccination"

_medicina, 2021, doi:10.3390/medicina57060611_

Round 1

Reviewer 1 Report

An interesting and relevant subject. It is an interesting, pretty straightforward text. The methodology is simple based on questionnaire collection with a descriptive statistics. Nice study with valid conclusions. A selection bias. as stated by authors. remains as a study limitation. And minor linguistic corrections needed. 

Author Response

Dear Editor,

                                                                                                                                     3 June 2021

Thank you for providing us the opportunity to submit a revised version of our manuscript. A point to point answer has been provided for the reviewers below. All authors have approved the revised version. I will serve as the corresponding author.

Thank you in advance for your kind consideration,

Best Regards,

Paul Zarogoulidis, M.D, Ph.D,

3rd Department of Surgery,

``AHEPA`` University Hospital,

Aristotle University of Thessaloniki,

Medical School,

Thessaloniki,

Greece

Mobile: 00306977271974

E-mail: pzarog@hotmail.com

We would like to thank the reviewers for their valuable comments and suggestions in order to improve our manuscript.

Reviewer 1

  • An interesting and relevant subject. It is an interesting, pretty straightforward text. The methodology is simple based on questionnaire collection with a descriptive statistics. Nice study with valid conclusions. A selection bias as stated by authors remains as a study limitation. And minor linguistic corrections needed.

Answer:

Thank you for your comments. We will make the appropriate linguistic corrections needed in order to improve our manuscript.

Reviewer 2

  • The study was of potential interest, but it results to be outdated at the present time. Nowadays vaccination campaigns have started in all the countries since some months ago, especially for HCPs.

Answer:

Thank you for your comments. The results seem outdated as vaccination campaigns have started in all the countries since some months ago; however  the problem on vaccine’s hesitancy exists even in HCPs. In the revised version of the manuscript we have updated the introduction and discussion with very recently published manuscripts addressing this problem.  The current survey has important implications as it is rather significant for each country to know the intention of HCPs to vaccinate as this is an indication of an important health policy maker for promoting the COVID-19 vaccination among the general population. We did not find many studies for the population of    Eastern Europe assessing this problem. Understanding healthcare workers’ vaccine hesitancy has substantial implications on public health administrations during epidemics.

  • Moreover, the information have not been updated (as an example, there is no mention about the approval of the J&J vaccine in Europe).

Answer: Thank you for your comment. We have included the approval of the J&J vaccine in Europe in the revised manuscript and also have revised appropriately the introduction and discussion.

  • Lines 90-91: the categories in the table seem to be three.

Answer: Thank you for your comment. In Table 2 we have included the results of several variables of the responders according to their intention to accept COVID-19 vaccination. In the regression analysis, we evaluated the factors associated with the responders’ intentions to accept COVID-19 vaccination.

  • Line 94: please indicate how many HCPs were invited, if reminders were sent and the response rate.

Answer:

Thank you for your comment. The HCPs were invited by the assistance of the local medical and nursing stuff associations. It is estimated that around 8,800 were invited. We did not sent reminders because we depended on the local associations that were not easy to cooperate at that time…Only 656 responded to the questionnaire (7.5%) (please see discussion in Limitations paragraph)

  • Moderate English review is required.

Answer: Thank you for your comment. We will make the appropriate linguistic corrections needed in order to improve our manuscript.

Reviewer 3

Introduction

  • The Authors should insert a point regarding the vaccine hesitancy since it gives a significant contribution to the attitudes and to the suboptimal vaccination coverage. The following articles should be cited Bianco et al. Vaccine 2019;37:984-90; Dubè et al. Hum Vaccin Immunother 2019;15:113-20; Napolitano et al. Hum Vaccin Immunother 2018;14:1558-65.

Answer: Thank you for your comments. We agree with your suggestion and have inserted a paragraph regarding vaccine hesitancy (please see revised manuscript)

‘The effectiveness of vaccines depends on their uptake by the population. Vaccine uptake may be influenced by concerns about their safety and effectiveness, low risk perception, lack of recommendation and information by health professionals and socio-economic factors.  Vaccine hesitancy constitutes a growing concern and includes the concerns people express by refusing or delaying vaccines despite their availability [3-5].  The World Health Organization (WHO) considered vaccine hesitancy to be one of the top ten health threats globally [6].’ 

Methods

  • It is indicated that a web-based survey was launched via Google but it is not indicated how many local medical and nursing stuff associations were contacted.

Answer: Thank you for your comment. The HCPs were invited by the assistance of the local medical and nursing stuff associations. The local associations are those of Thessaloniki in the Northern part of Greece (one medical and one of nursing staff). It is estimated that around 8,800 were invited.

  • The Authors should clarify how has been determined the sample size.

Answer: Thank you for your comment. The sample size was estimated with prevalence of participants willing to receive vaccination against COVID-19 set at 50%, a 95% confidence interval, and a relative precision of 5%. The estimated minimum sample size was 490.

  • The Authors should clarify about the face-validity testing of the questions with an explanation of the validity of the content of the questions with regard to the research aims. The Authors should clarify how they had estimated the reliability, or internal consistency, of the questions by using, for example the Cronbach’s alpha in order to measure whether or not a score is reliable.

Answer: Thank you for your comment. We could not apply Cronbach’s alpha in order to estimate the validity of the content of the questions because of the type of the questions. We used simple questions as: what is your age, what is your specialty, do you work in the ICU, What is your marital status, Do you have children, What is your financial status after the pandemic, Have you been infected from COVID 19, Do you treat patients with COVID 19.

We used only 4 Likert questions i.e. ‘Do you feel at risk of getting infected by COVID’, ‘Ηow venerable you consider yourself from COVID 19 infection?’, ‘If you were infected from COVID how serious  your condition would  be? rated on a 7-point scale and ‘wiliness  to get vaccinated  with COVID-19 vaccine’ rated on a 5-point scale. The estimated Cronbach’s alpha in ‘Do you feel at risk of getting infected by COVID’ was 0.87. , ‘Ηow venerable you consider yourself from COVID 19 infection?’ was 0.82, in ‘If you were infected from COVID how serious your condition would be?’ was  0.76 and in ‘wiliness  to get vaccinated  with COVID-19 vaccine’ was 0.94.

Additionally during a pilot study conducted before the actual data collection from a sample of 30 health care professionals working in our hospital (G Papanikolaou Hospital) we received very encouraging comments with no problems in the comprehension of the questions of the survey .

  • It is not stated whether the participant was informed about the use and anonymization of the data and that survey responses guarantee the anonymity of each participant

Answer: Thank you for your comment. An information sheet from the research team was included in the survey with a detailed explanation of the research purpose. The participants were informed about the use and anonymization of the data and that their anonymity would be perceived. We have added this to the revised manuscript.

  • It is not given any information whether participants were not able to continue to the next question of the questionnaire if they failed to provide a response to an item

Answer: Thank you for your comment. The participants were not able to continue to the next question of the questionnaire if they failed to provide a response to an item.

  • The Authors should clarify whether the participants have received any gift or monetarily compensated.

Answer: Thank you for your comment. The participants have not received any gift or monetarily compensated.

  • It should be clarified whether a pilot study has been conducted.

Answer:  A pilot study was conducted before the actual data collection from a sample of 30 health care professionals working in G Papanikolaou Hospital who completed the questionnaire and the results were not included in the survey. We have added this to the revised manuscript.

  • The Authors should describe the survey questionnaire items in more details.

Answer: Thank you for your comment. Information was collected through an anonymous online survey, structured questionnaire. An information sheet from the research team was included in the survey with a detailed explanation of the research purpose. The participants were informed about the use and anonymization of the data and that their anonymity would be perceived. No incentives were offered to the participants of the survey.

 The questionnaire addressed: (1) socio- demographical and general characteristics of the HCP (age, gender, marital and parenthood status, change of financial status during the pandemic),(2) occupation (physician, nurse, technician, or other healthcare professionals),  location of work (e.g. hospital, private practice), medical discipline (e.g. internal medicine, surgery, laboratory/imagine, intensive care unit), Years of experience, educational level(MD, phD), whether they directly provided care, diagnosed and treated patients with suspected or confirmed SARS-CoV-2, (3) the risk perception about COVID-19 was examined with the following  questions : ‘Do you feel at risk of getting infected by COVID’, “Ηow venerable you consider yourself from COVID 19 infection?” and  ‘If you were infected from COVID how serious  your condition would  be?, rated on a 7-point Likert scale with 1 indicating very mild to 7 indicating extremely (4) prior  infection from COVID 19 and (5)  wiliness  to get vaccinated  with COVID-19 vaccine with a five-point Likert scale (1 = strongly disagree, 2 = disagree, 3 = don’t know, 4 = agree,and 5 = strongly agree).

We have added this to the revised manuscript.

  • The statistical analysis is not adequate, because it would be particularly relevant to describe the multivariate model(s) developed and the outcome(s), and the rationale why the variables are included, and the model building strategy. It should be indicated the p-value that has been used to determine statistical significance and if the tests were one-side or two-sides.

Answer: Thank you for your comment.

Data were analyzed using IBM SPSS version 21.0 (IBM Corporation, Armonk, New York, USA).

 Continuous variables were presented as mean±SD unless otherwise stated. Absolute and relative frequencies were used for categorical variables. Analysis of variance (ANOVA) was used for the analysis of the differences between group means. Tests were two-tailed and p<0.05 was accepted as statistically significant. We combined survey responses into two categories (strongly agree/agree; don’t know/disagree/strongly disagree). Difference between groups with intention or not to accept vaccination and those undecided were examined using Chi-square test. Univariate logistic regression models were performed to assess relations between the different variables (gender, age, marital status, parenthood), specialty (surgical vs. internal medicine, laboratory vs. internal medicine), ICU work setting, type of HCP (i.e. physicians vs. nurses), work with confirmed /suspected COVID, infection  from COVID and the participants’ willingness to receive vaccination.

In order to isolate predictors of HCP willingness to get vaccinated against COVID-19, we performed a multivariate logistic regression. The following   independent variables were included: age in years (continuous), gender(male = 0; female = 1), marital (single/divorced/widowed = 0; married = 1) and parental status (no children =0, having children =1), type of healthcare worker(1=doctors, 2=nurses, 3 = others as technicians, laboratory, physiotherapists), specialty (internal medicine=1, surgical=2, laboratory/ imaging=3, ICU=4), treating patients with COVID 19 infection (no=0, yes =1) and being infected from COVID 19 themselves(no=0, yes =1) . The Odds Ratios (OR) are presented with a 95% confidence interval (CI).

We have added this to the revised manuscript.

Results

  • It is stated that the study population consisted of 656 HCPs, but no information has been given regarding how many individuals has been selected.

Answer: Thank you for your comment. The HCPs were invited by the assistance of the local medical and nursing stuff associations who sent the e-mails with the questionnaire to all their participants. It is estimated that around 8,800 were invited. However, only 656 of them answered the questionnaire. We have added this to the revised manuscript.

  • No information is given about how many individuals refused to participate and the response rate should be included (in the discussion it is stated that the response rate was rather low reaching 7.5%). If not all patients have agreed to participate, no information is given about them. Was there any attempt to quantify the response bias: information about non-responders. It would be useful to have some kind of indication of comparability with non-respondents. Is there any population-based data available? How did they differ from those in the sample, how representative is the sample and were the findings representative?

Answer: Thank you for your comment. We are not able to estimate the number of individuals that refused to respond. As the questionnaire was sent by the local associations to all their members, we can estimate a response rate from the number of members. That was how we reported the low response rate of 7.5%. On the other hand a limitation of the questionnaire was that the participants were not able to continue to the next question of the questionnaire if they failed to provide a response to an item, so some may have not completed the questionnaire and we do not have data from them. However, unofficially we have asked from the local associations to provide us data of the demographics of their participants and it seems that our data are rather representative for gender and age.  We have also added this to the limitations of the study in the revised manuscript.

Discussion

  • The discussion for large part is not appropriate. The major weakness is that the results are not compared with some of the studies conducted in other countries. The authors should expand their comments regarding the results of previous surveys conducted in different geographic areas regarding the willingness of receiving the vaccination against COVID-19 among healthcare workers. For example, the following article should be cited Di Giuseppe et al. Expert Rev Vaccines 2021 May 25:1-9. doi: 10.1080/14760584.2021.1922081; Kukreti et al. Vaccines (Basel) 2021;9(3):246; Verger et al. Euro Surveill 2021;26(3):2002047; Fisher et al. Ann Intern Med 2020;173(12):964-73.

Answer: Thank you for your comment as it has substantially improved the discussion of our revised manuscript. Please see the revised manuscript.

  • The paragraph regarding the main limitations of the study did not discuss all limits such as, for example, the study design, the method of sampling, the representativeness of the sample, the recall bias, and the social desirability bias.

Answer: Thank you for your comment. We have added these limitations to the revised manuscript.

References

  • The manuscript is not well referenced. The References list is not updated, since several articles conducted in different countries and published on peer-reviewed journals have been not included.

Answer: Thank you for your comment. We have added several new references. Please see the revised manuscript.

Tables

  • In Table, only one decimal should be reported.

Answer: Thank you for your comment. Please see the revised manuscript

  • Check that the total for each variable add to 100% (for example, in Table 1 Financial status after COVID 19 pandemic).

Answer: Thank you for your comment. Please see the revised manuscript

Reviewer 2 Report

The study was of potential interest, but it results to be outdated at the present time. Nowadays vaccination campaigns have started in all the countries since some months ago, especially for HCPs. 

Moreover, the information have not been updated (as an example, there is no mention about the approval of the J&J vaccine in Europe).

Lines 90-91: the categories in the table seem to be three.

Line 94: please indicate how many HCPs were invited, if reminders were sent and the response rate.

Moderate English review is required. 

Author Response

(The authors gave the same response as above.)

Reviewer 3 Report

Introduction

1. The Authors should insert a point regarding the vaccine hesitancy since it gives a significant contribution to the attitudes and to the suboptimal vaccination coverage. The following articles should be cited Bianco et al. Vaccine 2019;37:984-90; Dubè et al. Hum Vaccin Immunother 2019;15:113-20; Napolitano et al. Hum Vaccin Immunother 2018;14:1558-65.

Methods

  1. It is indicated that a web-based survey was launched via Google but it is not indicated how many local medical and nursing stuff associations were contacted.
  2. The Authors should clarify how has been determined the sample size.
  3. The Authors should clarify about the face-validity testing of the questions with an explanation of the validity of the content of the questions with regard to the research aims. The Authors should clarify how they had estimated the reliability, or internal consistency, of the questions by using, for example the Cronbach’s alpha in order to measure whether or not a score is reliable.
  4. It is not stated whether the participant was informed about the use and anonymization of the data and that survey responses guarantee the anonymity of each participant.
  5. It is not given any information whether participants were not able to continue to the next question of the questionnaire if they failed to provide a response to an item.
  6. The Authors should clarify whether the participants have received any gift or monetarily compensated.
  7. It should be clarified whether a pilot study has been conducted.
  8. The Authors should describe the survey questionnaire items in more details.
  9. The statistical analysis is not adequate, because it would be particularly relevant to describe the multivariate model(s) developed and the outcome(s), and the rationale why the variables are included, and the model building strategy. It should be indicated the p-value that has been used to determine statistical significance and if the tests were one-side or two-sides.

Results

  1. It is stated that The study population consisted of 656 HCPs, but no information has been given regarding how many individuals has been selected.
  2. No information is given about how many individuals refused to participate and the response rate should be included (in the discussion it is stated that the response rate was rather low reaching 7.5%). If not all patients have agreed to participate, no information is given about them. Was there any attempt to quantify the response bias: information about non-responders. It would be useful to have some kind of indication of comparability with non-respondents. Is there any population-based data available? How did they differ from those in the sample, how representative is the sample and were the findings representative?

Discussion

  1. The discussion for large part is not appropriate. The major weakness is that the results are not compared with some of the studies conducted in other countries. The authors should expand their comments regarding the results of previous surveys conducted in different geographic areas regarding the willingness of receiving the vaccination against COVID-19 among healthcare workers. For example, the following article should be cited Di Giuseppe et al. Expert Rev Vaccines 2021 May 25:1-9. doi: 10.1080/14760584.2021.1922081; Kukreti et al. Vaccines (Basel) 2021;9(3):246; Verger et al. Euro Surveill 2021;26(3):2002047; Fisher et al. Ann Intern Med 2020;173(12):964-73.
  2. The paragraph regarding the main limitations of the study did not discuss all limits such as, for example, the study design, the method of sampling, the representativeness of the sample, the recall bias, and the social desirability bias.

References

  1. The manuscript is not well referenced. The References list is not updated, since several articles conducted in different countries and published on peer-reviewed journals have been not included.

Tables

  1. In Table, only one decimal should be reported.
  2. Check that the total for each variable add to 100% (for example, in Table 1 Financial status after COVID 19 pandemic).

Author Response

(The authors gave the same response as above.)

Round 2

Reviewer 2 Report

The Authors have addressed the main concerns of the previous version of the manuscript.

Author Response

Thank you for all your help for improving our manuscript.

Reviewer 3 Report

Methods

  1. The fact that local associations are those of Thessaloniki in the Northern part of Greece (one medical and one of nursing staff) and the total number of affiliated should be indicated.
  2. Lines 88-89 and 345-347 It is stated that The participants were not able to continue to the next question of the questionnaire if they failed to provide a response to an item. However, in Table 1 for the variable Financial status after COVID 19 pandemic there 14.6% No answer. How is this possible?

Results

  1. No information is given about how many individuals refused to participate and the response rate should be included (in the discussion it is stated that the response rate was rather low reaching 7.5%).

Discussion

  1. The paragraph regarding the main limitations of the study still did not discuss adequately all limits such as the study design, the recall bias, and the social desirability bias.

Author Response

Thank you for all your help in order to improve our manuscript.

Methods

  • The fact that local associations are those of Thessaloniki in the Northern part of Greece (one medical and one of nursing staff) and the total number of affiliated should be indicated.

Answer: Thank you for your comment. The total number of affiliated was estimated around 8,800. Unfortunately only   656 of them answered the questionnaire. That is how we estimated that the response rate was 7.5%. We have added this information in the revised manuscript .

  • Lines 88-89 and 345-347 It is stated that The participants were not able to continue to the next question of the questionnaire if they failed to provide a response to an item. However, in Table 1 for the variable Financial status after COVID 19 pandemic there 14.6% No answer. How is this possible?

Answer: Thank you for your comment. There was an option in the question that the participant could choose ‘I do not answer’.

Results

  • No information is given about how many individuals refused to participate and the response rate should be included (in the discussion it is stated that the response rate was rather low reaching 7.5%).

Answer: Thank you for your comment. It is estimated that around 8,800 were invited (from the number of the members of the associations). Unfortunately only   656 of them answered the questionnaire. That is how we estimated that the response rate was 7.5%. We have added this information in the results section in the revised manuscript.

Discussion

  • The paragraph regarding the main limitations of the study still did not discuss adequately all limits such as the study design, the recall bias, and the social desirability bias.

Answer: Thank you for your comment. We have enriched the limitations section as you suggested in the revised manuscript.

‘Additionally, the representativeness of the sample, the recall bias and possible social desirability bias could be considered as limitations of the study.  At that time, northern Greece was more severely affected by COVID 19 compared to the other areas. The study design could be improved in a future project including a greater number of participants from more than one region (ideally national) and several associations, with a greater response rate of all the types of HCPs.  As human memory is frequently imprecise, recall bias could be included in the limitations of the study. Social desirability could also be considered a limitation, as the participants might have tended to over report the more socially desirable answers and to under report their actual attitudes’.
